**communications** engineering

# Solution-combustion synthesis of $Na_3(VO_{1-x})_2(PO_4)_2F_{1+2x}$ as a positive electrode material for sodium-ion batteries
Oskar Grabowski, Michal Krajewski ⊠, Magdalena Winkowska-Struzik & Andrzej Czerwinski

Sodium-vanadium fluorophosphates (NVPF) comprise a family of highly potent positive electrode materials for sodium-ion batteries, combining high energy density due to their high operating potential and good high-rate performance provided by NASICON structure. In this study, a self-combustion approach to synthesise $Na_3(VO_{1-x})_2(PO_4)_2F_{1+2x}$ ($0 \leq x \leq 1$) is reported. The method described here is based on a citrate-nitrate combustion process. As a result of the synthesis, phase-pure NVPF with uniform morphology and average grain size of 98 nm is obtained. Moreover, the self-combustion method implemented in this study results in the acquisition of a powder with excellent performance in Na-ion systems, showing high capacity (ca. 111 mAh g$^{-1}$), cyclability (ca. 94% of capacity retention), and high-rate performance (ca. 91% of capacity retention). The self-combustion technique described in this paper shows a promising approach to synthesising fluorinated polyanion compounds for Na-ion batteries.

Sodium-vanadium phosphate ($Na_3V_2(PO_4)_3$, NVP) is one of the most widely described positive electrode compounds for sodium-ion batteries. It crystallises in a NASICON structure, which provides good transport properties of sodium cations through a crystal framework and excellent structural, thermal, and chemical stability in Na-ion cells. However, due to the presence of heavy $PO_4^{3-}$ and $V^{4+}/V^{3+}$ redox centre, it is characterised by mediocre specific capacity (ca. 118 mAh g$^{-1}$) and operating potential (ca. 3.4 V vs. $Na^+/Na^0$) which result in low specific energy of the NVP material[1].

Increasing the redox potential of the $V^{4+}/V^{3+}$ centre can be realised through the introduction of fluoride anions into the crystal matrix, leading to the creation of a whole new family of compounds, described as $Na_3(VO_{1-x})_2(PO_4)_2F_{1+2x}$ ($0 \leq x \leq 1$). Due to the strong inductive effect of both F$^-$ and $PO_4^{3-}$ groups on the vanadium redox centre, the operating potential of such materials is increased when compared to pristine sodium-vanadium phosphate[2]. A fully fluorinated compound, sodium-vanadium fluorophosphate ($Na_3V_2(PO_4)_2F_3$, NVPF) is characterised by an operating potential of ca. 4.2 V vs. $Na^+/Na^0$ and a higher specific capacity of ca. 128 mAh g$^{-1}$ than non-fluorinated NVP[3]. The most commonly used synthetic approaches to obtain $Na_3V_2(PO_4)_2F_3$ are solid-state[2–9], sol-gel[10–13], electrospinning[14], and hydro- and solvothermal methods[15–24], which have been successfully adapted to synthesize NVPF with good electrochemical properties in Na-ion batteries.

One of the nanoparticle synthesis methods, that can rapidly produce uniformly distributed and small nanoparticles is combustion technique, utilising heat generated by exothermic reactions between oxidiser and fuel. This approach had been already successfully adapted to synthesise $Na_3V_2(PO_4)_3$, characterised by homogenously distributed nanoparticles[25,26]. However, the efforts to obtain fluorinated compounds from $Na_3(VO_{1-x})_2(PO_4)_2F_{1+2x}$ ($0 \leq x \leq 1$) family have not been described before.

In this study, we show a self-combustion synthetic approach to acquire sodium-vanadium fluorophosphate of NASICON structure. The use of this method resulted in synthesising phase-pure, uniform powder with excellent electrochemical properties in Na-ion systems.

## Results & Discussion
### Material Characterisation
X-ray diffraction (XRD) pattern obtained for the NVPF@C (NVPF/carbon) composite indicates that the solution combustion method is a successful procedure to synthesise $Na_3(VO_{1-x})_2(PO_4)_2F_{1+2x}$ materials (Fig. 1A). Obtained powder can be regarded as a solid solution of $Na_3V_2(PO_4)_2F_3$ and $Na_3(VO)_2(PO_4)_2F$ which can be rewritten as a $Na_3(VO_{1-x})_2(PO_4)_2F_{1+2x}$ ($0 \leq x \leq 1$). $Na_3V_2(PO_4)_2F_3$ structure is composed of bi-octahedral $V_2O_8F_3$ units linked by $PO_4$ groups (Fig. 1A, inset). Bi-octahedron consists of two octahedra with vanadium in each octahedron centre, four oxygen atoms in

Faculty of Chemistry, University of Warsaw, Pasteura 1, 02-093 Warsaw, Poland. ⊠e-mail: michal.krajewski@uw.edu.pl

**Fig. 1 | X-ray diffraction studies.** XRD pattern of single phase $Na_3(VO_{1-x})_2(PO_4)_2F_{1+2x}$ with the selected reflexes indexed[30] (**A**); the linear relationship between oxygen content in $Na_3(VO_{1-x})_2(PO_4)_2F_{1+2x}$ and c-parameter of the unit cell. Literature data are taken from Nguyen et al.[8]. **B** The error bars for the NVPF synthesised in this work (**B**) correspond to the uncertainty of c-parameter and $x$ values calculated through the propagation of uncertainty method.

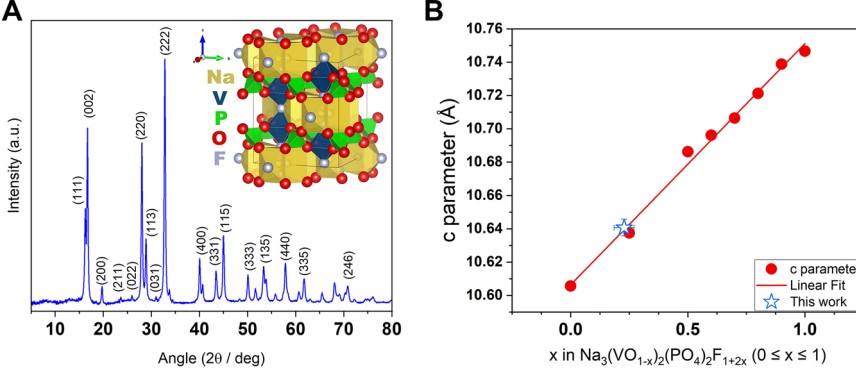

the *a-b* plane, and three fluorine atoms along the c-axis, one of them is shared between octahedra. Oxygen may partially substitute apical fluorine, inducing vanadium oxidation from 3+ to 4+ with the formation of a short covalent bond $V = O^{2,3,5,8,27-30}$. This leads to a slight distortion in bi-octahedra units and as a consequence compression of the unit cell along the c-axis. All distortions in a crystallography system manifest in a change of XRD patterns, thus it is possible to estimate chemical composition based on a c-axis parameter and volume per formula unit because a linear relationship was already described systematically[2,3,5,27,29,30]. Based on lattice parameters determined from Rietveld refinement (Supplementary Fig. 1, Supplementary Table 1), a self-combustion synthesis led to the preparation of a polyanionic compound with a composition close to $Na_3(VO_{0.77})_2(PO_4)_2F_{1.46}$ (Fig. 1B).

A linear approximation of the c-axis parameter as a function of oxygen content in $Na_3(VO_{1-x})_2(PO_4)_2F_{1+2x}$ reveals discrepancies across literature reports. Notably, for fully fluorinated compositions (x = 1), c-axis values range from 10.64 to 10.74 Å depending on the synthesis method, including solid-state, hydrothermal, combustion–thermal reaction (CTR), and sol–gel techniques[28,31-36]. A similar, though narrower, range is observed for oxidised analogues (x = 0), with reported c-axis values spanning 10.60–10.63 Å for materials synthesised via solid-state and solvothermal routes[30,31] (Supplementary Fig. 2). Across the full compositional range (0 ≤ x ≤ 1) slight deviations from linearity in c-axis evolution are evident, particularly between different synthetic approaches.

In principle, the relationship between the c-axis parameter, unit cell volume (V), or volume per formula unit (V/Z), and chemical composition is expected to follow Vegard's law – an empirical rule predicting a linear dependence of lattice constants on constituent element concentrations. For $Na_3(VO_{1-x})_2(PO_4)F_{1+2x}$, which crystallises in the tetragonal $P4_2/mnm$ space group, the highest c-axis values and cell volumes are anticipated for the fully fluorinated end-member (x = 1), while the lowest values are expected for the fully oxidised composition (x = 0). This trend is attributed to the difference in ionic radii between fluorine and oxygen, leading to contraction of the V–(O, F) bond lengths along the c-direction within the bi-octahedral $V_2O_8F_3$ units.

However, the observed variability in reported unit cell parameters, particularly at x = 1, raises questions regarding structural consistency. It remains unclear whether these discrepancies arise from unaccounted partial substitution of fluorine by oxygen, or from other structural deviations not explicitly addressed in prior studies.

SEM image of synthesized material is presented in Fig. 2A, B. It consists of fine particles with indeterminate shapes, ranging in size from about 50 to 200 nm, with a slight tendency to agglomerate into structures ca. 1 μm in diameter. However, the primary particles are mostly uniform, suggesting a good homogeneity of obtained powder through self-combustion technique. This might be the result of the extensive release of gasses during a burnout phase, which led to the spatial spacing of powder crystals, thus resulting in their reduced growth during the heat treatment process and later during synthesis[37,38]. In comparison, the morphology of NVPF synthesised via

various methods exhibits distinct and method-dependent characteristics. In the solid-state method, particles are predominantly agglomerated with irregular morphologies, spanning a broad size range from 200 nm to 5 μm, and no systematic trend in particle size or shape is observed with increasing oxygen content. Notably, this approach yields relatively large particles with a broad distribution[2,34,39]. The hydrothermal method results in particles with a well-defined cubic morphology, typically 2–3 μm in size, although noteworthy agglomeration is still evident[32]. For samples prepared by carbothermal reduction, SEM analysis reveals irregular particles ranging from 0.5 to 1 μm, with mechanochemical activation prior to sintering promoting the formation of larger secondary agglomerates[33]. In contrast, the solvothermal method yields more uniform micrometric aggregates composed of spherical primary particles with diameters between 200 and 400 nm, exhibiting a narrow and consistent size distribution across the sample[30].

HR-TEM imaging shown in Fig. 2C revealed well-crystallised NVPF particles (with an interplanar spacing of 0.536 nm originating from (002) crystal plane), coated with an amorphous carbon layer, in the form of onion-like carbon structures (Supplementary Fig. 3). Moreover, on the EDS elemental mapping image (Fig. 3), one can see uniformly distributed elements throughout the entire NVPF particles, which is in good agreement with XRD data and confirms the successful synthesis of phase-pure NVPF@C composite powder.

The $N_2$ adsorption/desorption experiment revealed that the adsorption isotherm shape follows a mixed IUPAC type-II/IV curves, suggesting that the analysed powder should be porous, with pores present mostly from the micro- and mesopore range[40,41]. The capillary condensation hysteresis present in Fig. 4A is a mix of IUPAC H2(b) and H3 hysteresis loops, suggesting that the microstructure of synthesized compound consists of pores with non-uniform shapes (but with a noteworthy presence of slit-like pores) and sizes. The pore distribution shown in Fig. 4B reveals, that the dominant pore widths are located in the 0.6–0.7, 3–4, and 9–10 nm ranges. A BET specific surface area calculated from the $N_2$ adsorption/desorption experiment reached $19.37 \pm 0.08$ m² g⁻¹, of which 3.32 m² g⁻¹ belongs to micropore specific surface area (calculated with the t-plot method). The obtained results are in good agreement with the scanning electron microscope (SEM) observations, showing a fine particle morphology, corresponding to a well-developed specific surface area of NVPF@C composite, due to the extensive evolution of gases during a burnout phase of the synthesis process. Moreover, the average grain size calculated from the adsorption/desorption experiment was found to be ca. 98 nm, which falls in the range of particle diameters observed on the SEM images.

The synthesis procedure was analysed using thermogravimetry. The initial mass loss, up to 300 °C, is mostly a result of adsorbed water removal from the surface of the compound and water particles remaining in the NVPF@C crystal structure[42]. Above 300 °C, oxidation of amorphous carbon begins and is continued up to 550 °C, with a corresponding mass loss of ca. 6.9% wt. Above 550 °C, one can see a gradual increase in the sample mass, most likely affiliated with the oxidation processes of the vanadium to the vanadium oxides (Fig. 5).

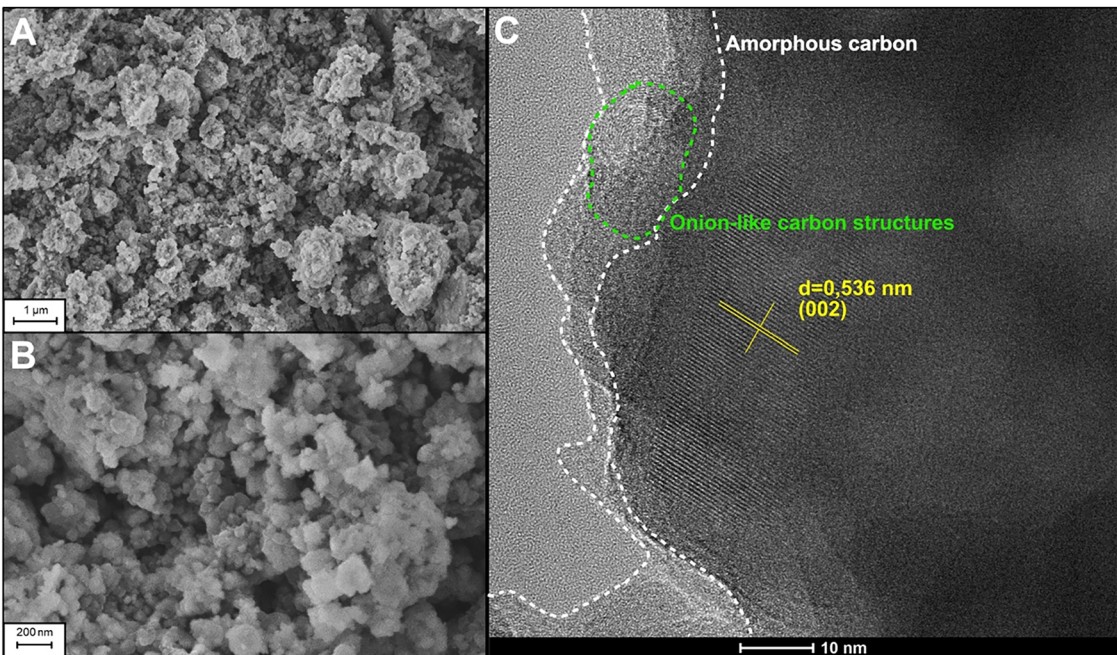

**Fig. 2 | Electron microscopy imaging results.** SEM images at 100,000 X (**A**) and 25,000 X (**B**) magnification and HR-TEM image (**C**) of NVPF@C composite powder.

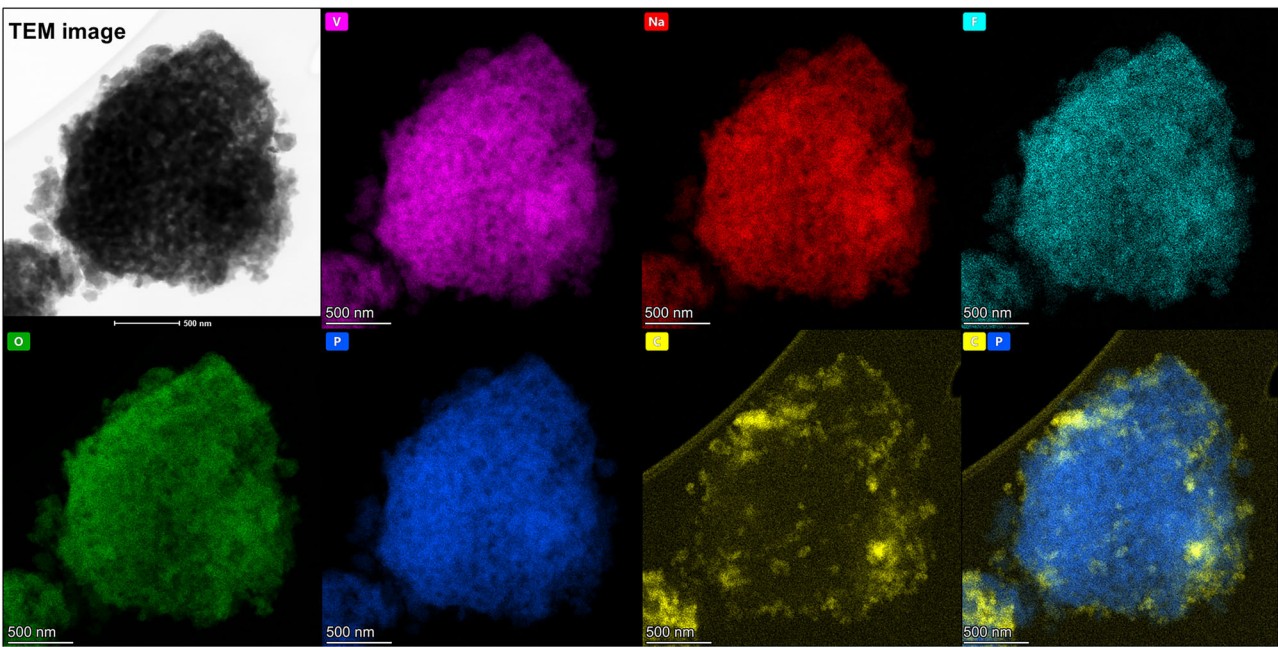

**Fig. 3 | Energy-dispersive X-ray spectroscopy mapping.** EDS mapping performed during the HR-TEM experiment on NVPF@C composite powder.

In general, the solution-combustion synthesis yielded phase-pure $Na_3(VO_{0.77})_2(PO_4)_2F_{1.46}$@C composite powder with uniform morphology and high specific surface area.

## Electrochemistry

**Chronopotentiometry.** Synthesised material provides a high discharge capacity of $110.4 \pm 0.3$ mAh g$^{-1}$ at 1 C discharge current concerning the mass of the whole NVPF@C composite (including mass of carbon coating present around NVPF particles after the synthesis). When the discharge current is raised, capacity drops negligibly to $108.3 \pm 0.2$, $105.2 \pm 0.1$, and $101.0 \pm 0.4$ mAh g$^{-1}$ at 2, 5 and 10 C, respectively (Fig. 6A). After the discharge current was lowered to 1 C for the second time, the discharge capacity increased to $108.76 \pm 0.22$ mAh g$^{-1}$ which

means, that the composite is appropriate for high-rate cycling and high-power applications. The results of the cyclability tests are shown in Fig. 6C. After 200 cycles of charging/discharging at 1 C current, NVPF@C retained $94.4 \pm 0.5\%$ of its initial capacity, showing excellent cyclability in Na-ion cells. When compared to sodium vanadium fluorophosphates obtained with other methods, NVPF@C composite synthesized through solution-combustion route shows superior rate performance and good long-term cycling stability[7,10,13,23,43–48] (Supplementary Table 2).

High magnification SEM images of the electrode cross sections before and after cyclability testing (Supplementary Fig. 4A, B) show that 200 charge and discharge cycles have not notably affected the morphology of the electrode content. Despite the extended operating time, there are no obvious

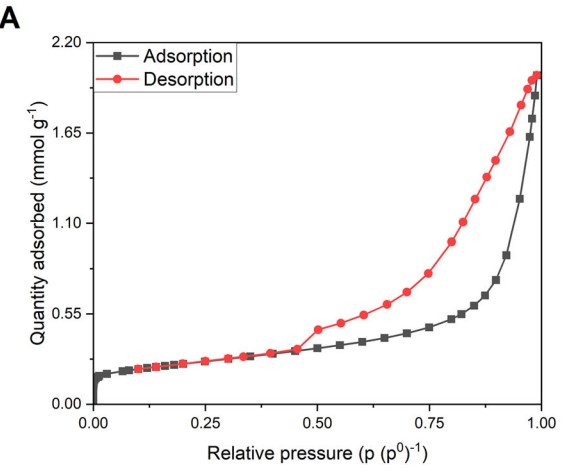

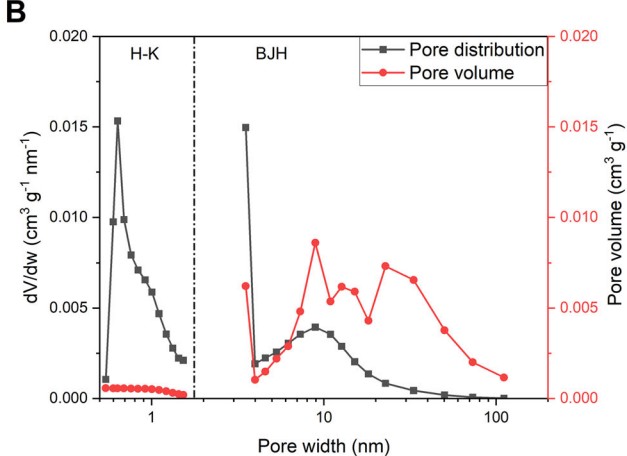

**Fig. 4 | N$_2$ adsorption/desorption experiments results.** Nitrogen adsorption/desorption isotherms (**A**) and pore distribution (**B**) of NVPF@C composite powder.

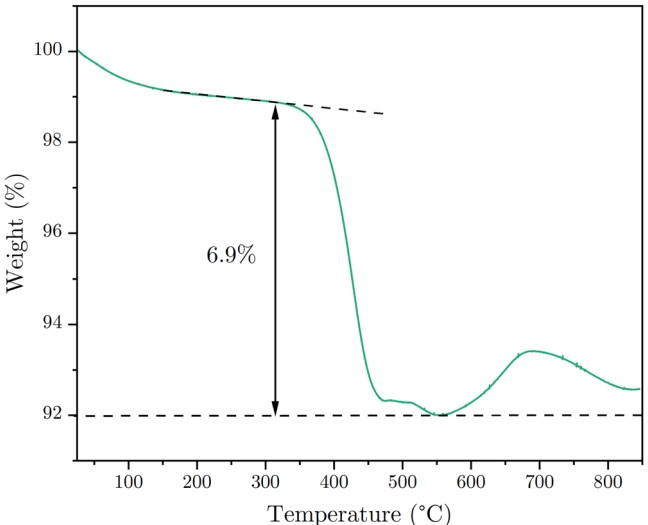

**Fig. 5 | Thermogravimetry analysis.** TGA curve measured for NVPF@C composite powder.

signs of damage, which is consistent with excellent capacity retention. Larger scale SEM images (Supplementary Fig. 4C, D) reveal the occurrence of partial delamination of the electrode mass after charge and discharge cycles, however this is more likely to be the result of cell disassembly and electrode cutting than an effect of prolonged cycling.

The XRD patterns of disassembled electrodes (Supplementary Fig. 5) correspond well with the patterns obtained for NVPF@C composite powder. Additional reflexes present on the diffractograms at 2Θ values of ca. 26°, 38°, 65°, and 78° were originating from other electrode components, namely Vulcan® XC72R (26°) and aluminium foil (38°, 65°, and 78°). No visible changes, between XRD patterns of the electrodes before and after 200 consecutive charge/discharge cycles, were observed, suggesting great structural stability and corresponds well with excellent cyclability of sodium-vanadium fluorophosphate obtained through solution-combustion method.

Two distinct potential plateaus at 3.6 and 4.05 V vs. Na$^+$/Na$^0$ are observed on charge and discharge curves (Fig. 6B, D) that are related to electrochemical intercalation/deintercalation of sodium ions into/out of the NVPF matrix. Those potentials do not coincide with theoretical values for the fully fluorinated Na$_3$V$_2$(PO$_4$)$_2$F$_3$ phase which are 3.7 and 4.2 V, presented earlier in the literature. A slightly lower value of the second potential plateau indicates that the obtained compound is not fully fluorinated and

the final formula is Na$_3$(VO$_{1-x}$)$_2$(PO$_4$)$_2$F$_{1+2x}$ with a notable $x$ value (as calculated from XRD data, $x = 0.229 \pm 0.042$). Due to the inductive effect caused by fluorine anions present in the crystal matrix, the average working potential of Na$_3$(VO$_{1-x}$)$_2$(PO$_4$)$_2$F$_{1+2x}$ compounds is increased linearly with $x$ value, as the valence states of some of the vanadium cations changes from 4+ to 3 +[2]. The average working potential in the case of NVPF@C synthesised through solution-combustion method was found to be 3.817 ± 0.039 V, which corresponds well with the literature data for $x = 0.229$[2]. The obtained powder is free from impurities since the voltage plateau at 3.4 V, related to sodium-vanadium phosphate (Na$_3$V$_2$(PO$_4$)$_3$), is not observed[49–51]. The galvanostatic charge/discharge cycling results are summarised in Table 1.

## Cyclic Voltammetry and Electrochemical Impedance Spectroscopy

Cyclic voltammograms of the NVPC@C compound revealed two, well-defined pairs of redox peaks, originating from mixed valence state V$^{5+}$/V$^{4+}$ (V$^{4+}$/V$^{3+}$) redox pairs oxidation/reduction processes during sodium extraction/insertion out/into the crystal matrix (Fig. 7A). The presence of two pairs of redox peaks is a result of changes in Na$^+$ chemical potential, due to the rearrangement of sodium ions in the crystal lattice after extraction of 1 mole of Na$^+$ per formula unit, to minimize the Na$^+$–Na$^+$ repulsion forces present in the crystal rather than due to the presence of a mixed valence state of vanadium cations in the crystal matrix[2]. Moreover, due to complicated phase transitions during NVPF oxidation/reduction reactions[52], additional signal separation can be seen on the voltammograms, mostly during reduction processes. Nevertheless, the average peak positions correspond well to the charge/discharge plateaus observed in Fig. 6B and 6D (Supplementary Tables 3 and 4). The Na$^+$ electrochemical diffusion coefficient values, calculated from the Randles-Ševčík relationship (Supplementary Fig. 7) were found to be $(1.62 \pm 0.02) \cdot 10^{-10}$, $(1.10 \pm 0.02) \cdot 10^{-10}$ cm$^2$ s$^{-1}$ for the 1$^{st}$ and $(1.92 \pm 0.02) \cdot 10^{-10}$, $(1.60 \pm 0.06) \cdot 10^{-10}$ cm$^2$ s$^{-1}$ for the 2$^{nd}$ oxidation/reduction processes, respectively. This suggests a slightly faster diffusion of sodium cations in a partially oxidised state of NVPF. Those values correspond well with the literature data for Na$^+$ diffusion coefficients in the sodium-vanadium fluorophosphate crystal matrix[4,53,54].

The EIS measurements were conducted to investigate the kinetic behaviour of NVPF@C composite powder. A complex plane plot of sodium-vanadium fluorophosphate in charged and discharged state is presented in Fig. 7B. In both cases, the plot consists of a depressed semicircle in high to mid frequencies and an inclined straight line in the low-frequency region. The equivalent circuit used for data fitting is shown in the inset of Fig. 7B. The intercept with the real axis at high frequency corresponds to ohmic and electrolyte resistances, followed by the semicircle related to charge-transfer resistance and the straight line originating from the

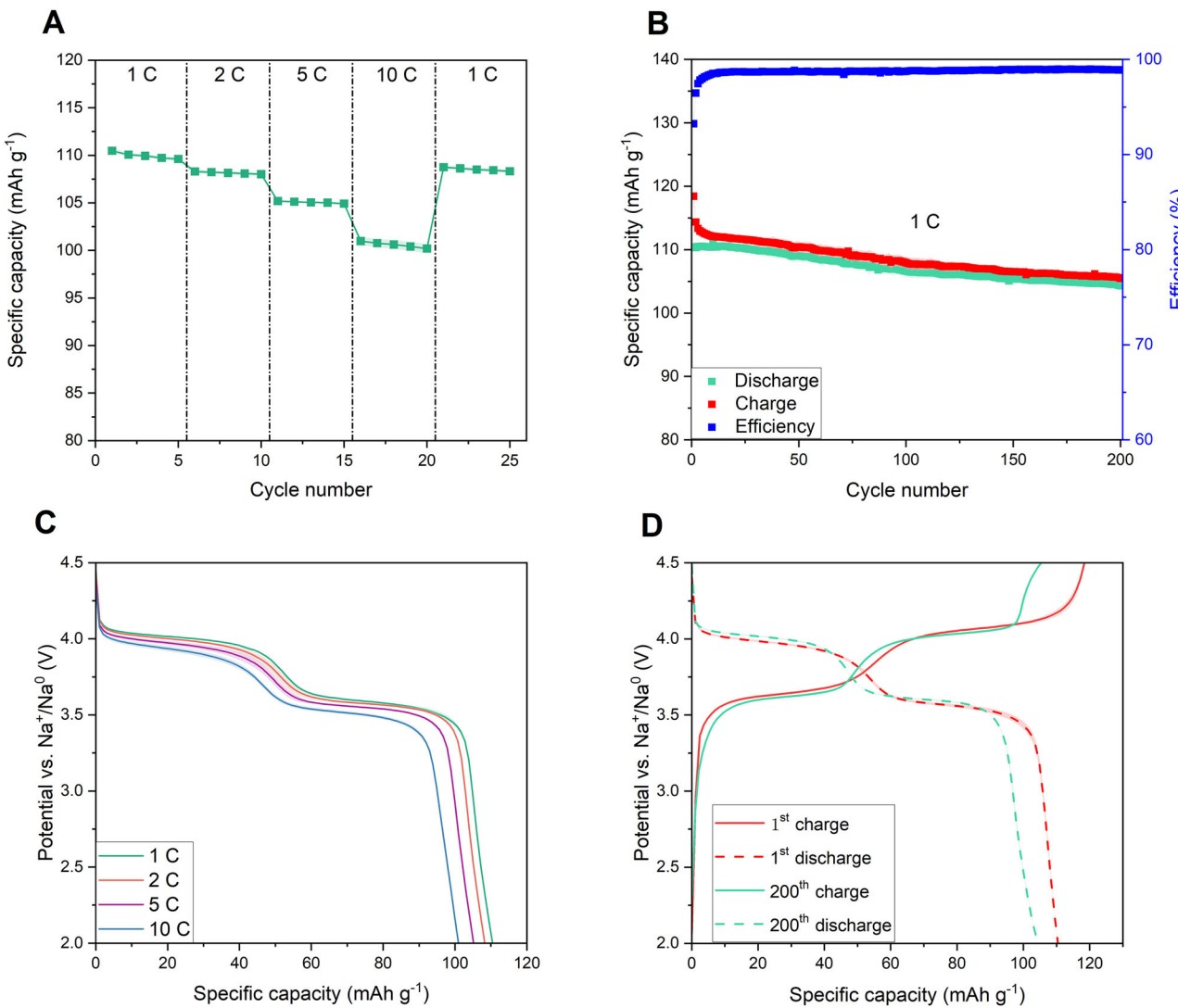

**Fig. 6 | Galvanostatic charging/discharging measurements.** High-rate performance (**A**), cyclability tests at 1 C current rate (**B**), discharge curves at 1, 2, 5, and 10 C discharge current rate (**C**), charge and discharge curves in 1st and 200th cycle (**D**) of NVPF@C. Blurred areas correspond to the standard deviation of the arithmetic mean acquired from NVPF@C performance between at least three different electrochemical cells.

### Table 1 | Results of Chronopotentiometry tests of NVPF@C compound in Na-ion cells

| 0.1 C (Supplementary Fig. 6) | 1 C cyclability tests | | |
|---|---|---|---|
| Preliminary charge/discharge capacity (mAh g$^{-1}$) | 1st cycle charge/discharge (mAh g$^{-1}$) | 200th cycle charge/discharge (mAh g$^{-1}$) | Capacity retained after 200 cycles (%) |
| 144.8 ± 4.4 / 114.1 ± 1.7 | 118.4 ± 0.6/110.4 ± 0.3 | 105.4 ± 0.8/104.3 ± 0.7 | 94.4 ± 0.5 |
| **High-rate tests** | | | |
| 1 C discharge capacity (mAh g$^{-1}$) | 2 C discharge capacity (mAh g$^{-1}$) | 5 C discharge capacity (mAh g$^{-1}$) | 10 C discharge capacity (mAh g$^{-1}$) | Capacity retained at 10 C rate (%) |
| 110.5 ± 0.3 | 108.3 ± 0.2 | 105.2 ± 0.1 | 101.0 ± 0.4 | 91.4 ± 0.6 |

capacitive effect of Warburg impedance and Na$^+$ diffusion. The calculated resistances from both spectra are included in Supplementary Table 5. One can see, that the charge-transfer resistance in the fully charged state is lower than in the reduced one, suggesting faster kinetics and electron transport in desodiated NVPF. This contributes well to CV findings, which showed faster Na$^+$ transport properties in the oxidised state of sodium-vanadium fluorophosphate. However, due to similarities between time constants for charge transfer and diffusion processes, we could not calculate the Na$^+$ electrochemical diffusion coefficient from EIS data, due to the penetration of the AC signal into the centre of the NVPF particles (reflected by a vertical capacitive line on the complex-plane plots)[55,56], which could be expected from a compound of NASICON structure showing fast ionic transport properties.

The Na$_3$(VO$_{0.77}$)$_2$(PO$_4$)$_2$F$_{1.46}$ obtained through solution-combustion method was, in general, characterized by excellent cyclability and high-rate performance in Na-ion systems.

## Conclusions

We have successfully synthesised sodium-vanadium fluorophosphate through the solution-combustion method. The obtained powder of

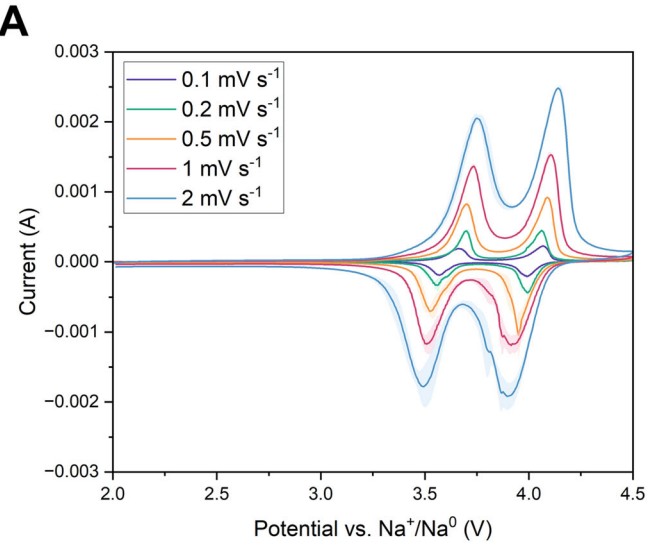
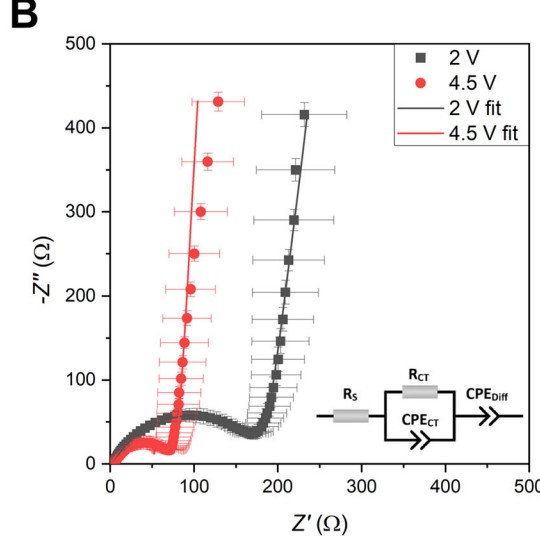

**Fig. 7 | Kinetic evaluation results.** Cyclic voltammograms at various scan rates scan rates (**A**) and complex-plane plots (**B**) of NVPF@C electrodes in Na-ion cells. The inset in Fig. 4B shows the equivalent circuit used for data fitting. Blurred areas and error bars correspond to the standard deviation of the arithmetic mean acquired from NVPF@C performance between at least three different electrochemical cells.

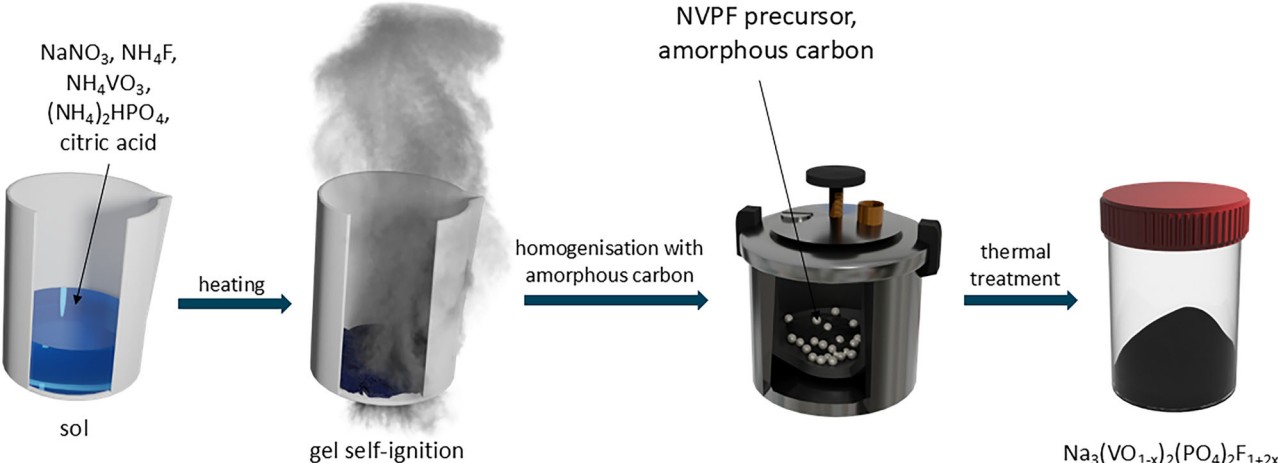

**Fig. 8 | Self-combustion synthesis diagram.** Schematic representation of self-combustion synthesis of $Na_3(VO_{1-x})_2(PO_4)_2F_{1+2x}$.

NASICON structure was a phase-pure and did not contain any traces of the most common impurity in this class of compounds, mainly $Na_3V_2(PO_4)_3$. SEM imaging showed a powder, with uniform morphology and $N_2$ adsorption/desorption revealed its well-developed specific surface area and average grain size in good agreement with SEM imaging. HR-TEM/EDS experiments revealed a uniform distribution of elements in the powder grains and the presence of carbon-coating surrounding NVPF particles. Galvanostatic charge/discharge cycling showed excellent performance of synthesized material, with cyclability reaching $94.4 \pm 0.5\%$ capacity retained after 200 consecutive charge/discharge cycles and exceptional high-rate performance, retaining $91.4 \pm 0.6\%$ of specific capacity, after changing the discharge current from 1 to 10 C rate. Although synthesis of the fully-fluorinated compound of $Na_3V_2(PO_4)_2F_3$ was not achieved in this study, we show that the self-combustion approach can be a useful method to synthesize materials of NVPF family, which is characterised by very good electrochemical performance.

Our future work will focus on synthesising $Na_3(VO_{1-x})_2(PO_4)_2F_{1+2x}$ ($0 \le x \le 1$) through the entire $x$ range and optimise the synthesis parameters to obtain an excellent positive electrode material for use in new generation sodium-ion batteries.

## Materials & Methods
### Synthesis[57]
Stoichiometric amounts of $NH_4F$ (Sigma-Aldrich), $(NH_4)_2HPO_4$ (Chempur), and $NaNO_3$ (Chempur) were firstly dissolved in 50 ml of deionized water in a Teflon® beaker (solution 1). Simultaneously, stoichiometric amounts of $NH_4VO_3$ (Sigma-Aldrich) and citric acid (POCH) (the molar ratio of citric acid to vanadium ions was equal to 2:3) were dissolved in 50 ml of deionized water in a glass beaker (solution 2). Both solutions were subject to magnetic stirring and heating at 80 °C. After solution 2 changed colour to blue, it was added drop by drop to solution 1 and continuously stirred and heated until complete evaporation of the solvent. Afterward, the remaining precipitate was heated to 150 °C, initiating self-combustion. The so-obtained green powder was dried in a vacuum at 120 °C for 12 h and then homogenized with Vulcan® XC72R (Cabot) carbon (powder: carbon ratio of 14:1 wt./wt.) in Fritsch Pulverisette 7 Premium Line planetary ball mill at 500 rpm for 5 h. This process was conducted in a tungsten carbide grinding vial with 250 ceramic balls. Lastly, the powder was heat-treated in a tube furnace at 300 °C for 4 h and then at 550 °C for 8 h in an Ar atmosphere. A schematic representation of the synthesis procedure is presented in Fig. 8.

## Material Characterisation

The crystal structure and phase composition of the samples were determined by X-ray diffraction (XRD) analysis. XRD patterns were collected using a Panalytical Empyrean diffractometer with Cu $K_\alpha$ radiation ($K_{\alpha 1}$ = 1.54060 Å, $K_{\alpha 2}$ = 1.54443 Å) in the 2θ range of 5° to 115° with a Bragg-Brentano geometry. The scan speed was set at 250 cps, and the step size was 0.02°. The obtained data were analysed using HighScore 4.0, Match!®, and FullProf software to identify the sample's phases and calculate structural parameters using Rietveld refinement. The structural data were evaluated for three different powders acquired through the synthetic approach explained above. SEM images were obtained through a Merlin (Zeiss) microscope using a 3 kV electron beam. TEM imaging was conducted using TALOS F200X (Thermo Scientific) microscope with state-of-the-art Energy Dispersive X-ray Spectroscopy (EDS) signal detection and 3D chemical characterisation with compositional mapping. $N_2$ adsorption/desorption experiments were conducted on ASAP 2060 (Micromeritics®) at 77.349 K in the relative pressure range of 0.01–0.995 $p \, (p^0)^{-1}$. Microporosity of NVPF powder was analysed through the insertion of fixed amounts of $N_2$ up to a relative pressure of 0.01 $p \, (p^0)^{-1}$. BET specific surface area, micropore specific surface area, and average grain size (assuming ideally spherical particle geometry and density of NVPF crystal of 3.167 g cm$^{-3}$) were calculated through ASAP 2060 software. Pore distribution was calculated through the BJH (Barrett-Joyner-Halenda) model in the mesopore range and the Horváth-Kawazoe (H-K) method in the micropore range. TGA analysis was conducted on a Q50-1091 thermogravimetric analyser (TA Instruments) in the temperature range from room temperature to 850 °C, with a heating ramp of 10 °C min$^{-1}$ and under $O_2$ flow of 5 ml min$^{-1}$.

## Electrode and Cell Preparation

$Na_3(VO_{1-x})_2(PO_4)_2F_{1+2x}$ powder was firstly mixed in an agate mortar with Vulcan® XC72R (Cabot) conductive carbon for 20 min. Afterward, 5% wt. PVDF (Alfa Aesar) solution in NMP (Sigma-Aldrich) was added to the powder, and the obtained mixture was then homogenized with centrifugal mixer SpeedMixer® DAC 150.3 FVZ (Hauschild) in two 5 min cycles at 3000 rpm. The ratio of NVPF:PVDF: Vulcan® was 8:1:1 wt. %. The obtained slurry was then applied to the surface of aluminium foil by a doctor blade, dried at 55 °C in the air to remove NMP, and further dried in a vacuum at 120 °C overnight. Subsequently, round, 9 mm in diameter, electrodes were cut from the foil, pressed under a hydraulic press at 6 t for 15 s, thoroughly weighed, dried at 120 °C in a vacuum overnight, and transferred to an argon-filled glove box (MBraun), with $H_2O$ and $O_2$ contamination level below 0.5 ppm.

The electrochemical cells were constructed in Swagelok® three-electrode geometry, with working electrode made of NVPF (electrode loadings of 2.69 ± 0.08 and 2.05 ± 0.02 mg cm$^{-2}$ during cyclability and high-rate/CV-EIS tests, respectively), counter and reference electrodes made of metallic sodium (Sigma-Aldrich), and Whatman® GF/F separator soaked in 1 M $NaClO_4$ (Sigma-Aldrich) solution in ethylene carbonate/propylene carbonate (1:1 v/v, Sigma-Aldrich).

## Chronopotentiometry

Galvanostatic charge/discharge cycling was performed on multichannel battery tester ATLAS 0961 (Sollich) in the potential range of 2-4.5 V vs. Na$^+$/Na$^0$. Cells were subject to cyclability tests and high-rate tests in which the first preliminary cycle was conducted at 0.1 C current rate (where C corresponds to 128 mA g$^{-1}$).

After cell conditioning, during cyclability evaluation, they were charged/discharged at a 1 C rate for 200 consecutive cycles. After testing cells were disassembled in glovebox under Ar atmosphere. Electrodes were extracted, washed several times with propylene carbonate and dried overnight. As-prepared samples were submitted to XRD analysis. In order to examine the morphology in cross section with SEM imagining electrodes were cut in half with precise scissors. For comparison, the pristine electrodes were also subjected to both XRD and SEM analysis. During high-rate tests, cells were charged at 1 C rate and discharged at various current rates, mainly 1, 2, 5, 10, and 1 C for 5 consecutive cycles, respectively.

## Cyclic Voltammetry

Straight after high-rate tests, cells were subject to CV analysis, conducted at SI 1287 Electrochemical Interface (Solartron) in the potential range of 2.00-4.50 V vs. Na$^+$/Na$^0$. The working electrodes were polarized with a scan rate of 0.1, 0.2, 0.5, 1.0 and 2.0 mV s$^{-1}$. For the sodium ion diffusion coefficient ($D$, cm$^2$ s$^{-1}$) calculations through Randles-Ševčík equation:

$$I_\mathrm{p} = 0.4463 \left( \frac{n^3 F^3}{RT} \nu D \right)^{1/2} AC,\tag{1}$$

in which $\nu$ is scan rate (V s$^{-1}$), $n$ is number of electrons involved during the electrochemical process, $F$ is Faraday's constant (C mol$^{-1}$), $R$ is gas constant (J (mol K)$^{-1}$), $T$ is absolute temperature (K) – the geometric surface area of the electrodes ($A$, cm$^2$) and Na$^+$ concentration ($C$) of 0.023 mol cm$^{-3}$ were used.

## Electrochemical Impedance Spectroscopy

Cells subjected to high-rate cycling and CV analysis were examined by EIS measurements performed on SI 1287 Electrochemical Interface (Solartron) with SI 1260 Impedance/Gain-Phase Analyzer (Solartron). The EIS experiments were conducted at potentials of 2.00 and 4.50 V vs. Na$^+$/Na$^0$ (after one hour of cell conditioning), with the AC amplitude of 5 mV (RMS) in $10^{-1}$–$10^4$ Hz frequency range. The EIS spectra were analysed using RelaxIS 3 software.

## Data availability

The datasets generated and/or analysed during the current study are available in the "Dane Badawcze UW" repository, https://doi.org/10.58132/7PG6LT.

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

## Acknowledgements

This work was supported by The Polish National Centre of Science through a research grant 2021/43/D/ST5/01220 entitled "High-voltage sodium-ion batteries based on exfoliated graphite electrodes".

## Author contributions

O.G. optimised synthesis parameters, carried out microscopy, thermogravimetric, galvanostatic and cyclic voltammetry evaluation and was responsible for writing the original draft. M.K. carried out $N_2$ adsorption/desorption and electrochemical impedance spectroscopy investigation, gathered funding and was responsible for conceptualisation, methodology and results verification. M.W.-S. carried out structural investigation. O.G and M.W.-S. carried out the synthesis process. M.K. and A.C. supervised the research. All authors contributed equally to review and editing of the manuscript.

## Competing interests

The authors declare no competing interests.
