## [Transparent Peer Review file · Communications Engineering]

Solution-combustion synthesis of $\text{Na}_3(\text{VO}_{1-x})_2(\text{PO}_4)_2\text{F}_{1+2x}$ as a positive electrode material for sodium-ion batteries

Corresponding Author: Dr Michal Krajewski

Version 0:

Reviewer comments:

Reviewer #1

(Remarks to the Author)

- 1- What is the motivation behind this work?
- 2- Why is the benefit of using self-combustion synthetic approach comparing with previously used approaches?
- 3- Please modify punctuation and needed superscripts throughout the paper.
- 4- Please fix the formatting of tables in the manuscript and supplementary information sheet
- 5- It is mentioned that "the straight line originating from the capacitive effect of Warburg impedance" however it is not matching with the represented equivalent circuit used for data fitting shown in the inset of Figure 7B.

Reviewer #2

(Remarks to the Author)

The manuscript presents a comprehensive study on the solution-combustion synthesis of $\text{Na}_3(\text{VO}_{1-x})_2(\text{PO}_4)_2\text{F}_{1+2x}$ as a positive electrode material for sodium-ion batteries. Here are some comments and suggestions for improvement:

Abstract:The abstract is well-written but could be more concise. Consider summarizing the key findings in fewer words to make it more impactful.

Introduction:The introduction provides a good background but could benefit from a clearer statement of the research gap and objectives. Explicitly state what makes the self-combustion method novel compared to existing methods.

Results & Discussion:The results section is detailed and thorough. However, it might be helpful to include a brief summary at the end of each subsection to highlight the key findings.

In the XRD and SEM sections, consider adding more comparative analysis with existing literature to emphasize the advantages of the self-combustion method.

The electrochemical performance data is well-presented. Including a comparison table with other synthesis methods could further highlight the benefits of the self-combustion approach.

Figures and Tables:Ensure all figures and tables are clearly labeled and referenced in the text. Some figures, like Figure 1, could benefit from more detailed captions explaining the key observations. The table summarizing the chronopotentiometry tests is informative but could be formatted for better readability on mobile devices.

Materials & Methods: The synthesis procedure is detailed, but a flowchart or schematic (like Figure 8) at the beginning of this section could help readers quickly grasp the process.

Consider providing more details on the reproducibility of the synthesis method and any potential challenges.

Conclusion: The conclusion effectively summarizes the findings but could be more forward-looking. Discuss potential future work or applications of the synthesized material.

Overall, the manuscript is well-structured and presents significant findings. Addressing these comments could enhance the clarity and impact of the study.

Reviewer #3

(Remarks to the Author)

The manuscript describes a novel solution-combustion method for synthesising $\text{Na}_3(\text{VO}_{1-x})_2(\text{PO}_4)_2\text{F}_{1+2x}$ (where $0 \leq x \leq 1$) as a positive electrode material for Na-ion batteries. The material demonstrated high capacity retention and exceptional high rate performance demonstrated at a 10 C discharge rate. The study highlights the method's potential for producing high performance materials in Na-ion battery systems, despite the compound not achieving full fluorination. The reviewer recommends this manuscript for publication, only after the authors addressed the aforementioned queries and recommendations to enhance clarity and impact:

1. Line 46: The statement that this solution combustion method can successfully be used to synthesise NASICON compounds is too general of a claim. It is recommended that the authors only limit the claim to the material reported in this manuscript.
2. Line 71: The claim that a good homogeneity of the primary particle size formed is due to the extensive release of gasses during the burnout phase, requires literature support. Otherwise, this statement is a pure hypothetical claim from the authors, unless the authors can provide analytical proof.
3. Line 75: Check the figure referencing of Figure S3.
4. Line 76: "Onion-like carbon structures" is not commonly used in the literature to describe amorphous carbon coating around the particle. The authors should elaborate more on this.
5. Figure 4: The lines of both the plots can be thicker to enhance legibility and clarify of the data.
6. Line 104: The removal of surface moisture during TGA will be complete at around 100 deg C. Only certain types of moisture (e.g. absorbed moisture or water molecules tightly associated with the material) may not fully evaporate until higher temperatures are reached. Therefore, the authors will need to justify why the mass loss up to 300 deg C represents loss of surface moisture.
7. Line 115: Please elaborate the meaning of "concerning the mass of the whole NVPF@C composite".
8. In the main text, the result in Figure 6C is discussed before Figure 6B. This is not a serious issue, but to promote ease of reading, it is recommended to rearrange the figures.
9. Line 123: Two distinct voltage plateaus at 3.6 and 4.05 V are observed in the voltage profile of the synthesised NVPF. Since the authors have claimed that these plateaus do not correspond to pure NVPF (3.7 and 4.2 V), and there is no plateau at 3.4 V which corresponds to NVP, the reviewer fails to understand the significance of the plateaus at 3.6 and 4.05 V. There is a huge mismatch in position the voltage plateaus.
10. Also to avoid any ambiguity of the origin of the observed voltage profiles, it is suggested that the authors conduct an ex-situ XPS analysis on the cycled NVPF electrode (refer to Section 3.6 of this paper: <https://www.sciencedirect.com/science/article/abs/pii/S2405829717301423>).
11. Line 164: Instead of using "oxidised state", the more commonly used term is "fully charged state".
12. The Na chemical diffusion coefficient can be easily determined from the low frequency EIS plot (Warburg contribution). Please refer to these papers: <https://link.springer.com/article/10.1007/s10008-011-1426-4> and <https://www.sciencedirect.com/science/article/abs/pii/S1388248108003469>.
13. Line 211: This section can be conveniently named Material Characterisations.
14. Line 245: Double check if the electrode mass loading unit is g/cm² or mg/cm².
15. Since the authors report a NVPF@C composite, it is recommended that a TGA is performed to quantify the amount of carbon present in the composite.
16. Standardise the use of "Na⁺ ion", "Na⁺ cation" and "Na⁺". The correct representation should be either "Na⁺" or "Na ion". Since Na⁺ is already stands for sodium ion, writing Na⁺ ion is redundant.
17. The authors should review the language and tenses throughout the manuscript.

Version 1:

Reviewer comments:

Reviewer #1

(Remarks to the Author)

The authors have addressed the reviewer comments, and the article can now be accepted for publication.

Reviewer #2

(Remarks to the Author)

I have reviewed the changes and I am happy to recommend the publication of this manuscript.

Reviewer #3

(Remarks to the Author)

I have reviewed the authors' responses and the revised manuscript. I appreciate the thoughtful and thorough manner in which all comments and feedback are addressed.

The revisions have significantly improved the clarity and quality of the manuscript. Hence, I have no further concerns and am satisfied with the current version.

Dear Reviewers,

we thank you for your feedback regarding our Manuscript. We attached our responses for your questions below.

With my best regards,

Dr. Michal Krajewski on behalf of the authors.

General comments:

1. Provide XPS analysis of the cathode materials.

The XPS analysis of $\text{Na}_3(\text{VO}_{1-x})_2(\text{PO}_4)_2\text{F}_{1+2x}$ ($0 \leq x \leq 1$) compounds will be performed in our next research topic, focusing on optimisation of the solution-combustion method to obtain $\text{Na}_3(\text{VO}_{1-x})_2(\text{PO}_4)_2\text{F}_{1+2x}$ ($0 \leq x \leq 1$) compounds in the entire range of x .

2. Include long-term cycling to see the stability of the cathode materials.

We included long-term cycling of NVPF@C electrodes in the Manuscript.

3. Post-mortem analysis which includes, XRD, SEM and XPS after the cycling.

We included post-mortem analysis of NVPF@C electrodes after 200 consecutive charge/discharge cycles, which showed no visible changes in SEM and XRD data after prolonged cycling.

Reviewers' comments:

Reviewer #1 (Remarks to the Author):

1- What is the motivation behind this work?

Solution-combustion method have been already successfully adopted to synthesise sodium-vanadium phosphate (NVP). However, sodium-vanadium fluorophosphate (NVPF), the fluorinated NVP have been characterised by higher operating potential, which can be a crucial parameter in Na-ion batteries, due to 0.3 V difference between Li^+/Li^0 and Na^+/Na^0 redox potentials. To the best of our knowledge, the solution-combustion approach to synthesise fluorinated compounds from $\text{Na}_3(\text{VO}_{1-x})_2(\text{PO}_4)_2\text{F}_{1+2x}$ ($0 \leq x \leq 1$) family have not been described before, which prompted us to develop the synthesis method described in the Manuscript.

We included additional information about the motivation behind our in the Manuscript.

2- Why is the benefit of using self-combustion synthetic approach comparing with previously used approaches?

According to the literature, the combustion techniques can rapidly produce uniformly distributed and small nanoparticles, which is beneficial for electrochemical performance of powder materials in Na-ion batteries.

We added additional information and references about combustion method into the Manuscript.

3- Please modify punctuation and needed superscripts throughout the paper.

We reviewed the language and punctuations throughout the Manuscript.

4- Please fix the formatting of tables in the manuscript and supplementary information sheet

The formatting of the tables were checked and fixed for better readability.

5- It is mentioned that “the straight line originating from the capacitive effect of Warburg impedance” however it is not matching with the represented equivalent circuit used for data fitting shown in the inset of Figure 7B.

The impedance response from NVPF@C electrodes at mid to low frequencies consists of two processes – charge transfer through electrode/electrolyte interphase and Warburg diffusion of Na⁺ through NVPF crystal. The diffusion of sodium cations in NVPF matrix is considered fast (NVPF crystallises in NASICON structure) and, due to the presence of homogenous, small NVPF particles (ca. 98 nm as calculated from BET) in the electrode, both processes cannot be distinguished from each other on the complex-plane plots. In that case, at mid to low frequencies, the AC signal can easily penetrate into the centre of the particles and, instead of producing a characteristic straight line inclined at 45° at complex-plane plots, it gives almost vertical, capacitive response (the Na⁺ cannot penetrate deeper than the centre of the particle, resulting in capacitive-like response). Trying to fit the data with an equivalent circuit which includes Warburg element will produce unsatisfactory results, due to software trying to fit the data into the diffusion equations first (which result in construction of the 45° slope). Using constant-phase element instead, one can get better fitting results, in the case of charge-transfer and diffusion time constants being of similar values, which is the case in our system. We added additional information and references about EIS data into the Manuscript.

Reviewer #2 (Remarks to the Author):

The manuscript presents a comprehensive study on the solution-combustion synthesis of Na₃(VO_{1-x})₂(PO₄)₂F_{1+2x} as a positive electrode material for sodium-ion batteries. Here are some comments and suggestions for improvement:

Abstract: The abstract is well-written but could be more concise. Consider summarizing the key findings in fewer words to make it more impactful.

We included key findings and performance parameters in the Abstract.

Introduction: The introduction provides a good background but could benefit from a clearer statement of the research gap and objectives. Explicitly state what makes the self-combustion method novel compared to existing methods.

We added additional information about combustion method into the Manuscript.

Results & Discussion: The results section is detailed and thorough. However, it might be helpful to include a brief summary at the end of each subsection to highlight the key findings.

We added a brief summary at the end of each section in the Manuscript.

In the XRD and SEM sections, consider adding more comparative analysis with existing literature to emphasize the advantages of the self-combustion method.

We have added a comparative analysis of synthetic procedures regarding XRD and SEM of NVPF family materials.

The electrochemical performance data is well-presented. Including a comparison table with other synthesis methods could further highlight the benefits of the self-combustion approach.

We included a comparison table and added it into the Supporting Information (Table S2). We also made a brief comparison about synthetic procedures and placed it into the Manuscript.

Figures and Tables: Ensure all figures and tables are clearly labeled and referenced in the text. Some figures, like Figure 1, could benefit from more detailed captions explaining the key observations. The table summarizing the chronopotentiometry tests is informative but could be formatted for better readability on mobile devices.

We changed the figure's labels and references in the text. Structural data is discussed in more detail in the Manuscript and Supplementary Information. Table 1 was also reformatted for better readability.

Materials & Methods: The synthesis procedure is detailed, but a flowchart or schematic (like Figure 8) at the beginning of this section could help readers quickly grasp the process.

We changed the Figure 8 placement in the text.

Consider providing more details on the reproducibility of the synthesis method and any potential challenges.

The synthesis of NVPF@C were repeated three times. All the structural data, calculations and so on were compared with each other. The calculations of the uncertainty of structural parameters and fluorine calculations were also performed. We added the information about reproducibility in the Materials & Methods section.

Conclusion: The conclusion effectively summarizes the findings but could be more forward-looking. Discuss potential future work or applications of the synthesized material.

We added additional information about the future work and application of the synthesised material.

Overall, the manuscript is well-structured and presents significant findings. Addressing these comments could enhance the clarity and impact of the study.

Reviewer #3 (Remarks to the Author):

The manuscript describes a novel solution-combustion method for synthesising $\text{Na}_3(\text{VO}_{1-x})_2(\text{PO}_4)_2\text{F}_{1+2x}$ (where $0 \leq x \leq 1$) as a positive electrode material for Na-ion batteries. The material demonstrated high capacity retention and exceptional high rate performance demonstrated at a 10 C discharge rate. The study highlights the method's potential for producing high performance materials in Na-ion battery systems, despite the compound not achieving full fluorination. The reviewer recommends this manuscript for publication, only after the authors addressed the aforementioned queries and recommendations to enhance clarity and impact:

1. Line 46: The statement that this solution combustion method can successfully be used to synthesise NASICON compounds is too general of a claim. It is recommended that the authors only limit the claim to the material reported in this manuscript.

We changed our statement to only refer compounds from $\text{Na}_3(\text{VO}_{1-x})_2(\text{PO}_4)_2\text{F}_{1+2x}$ ($0 \leq x \leq 1$) family.

2. Line 71: The claim that a good homogeneity of the primary particle size formed is due to the extensive release of gasses during the burnout phase, requires literature support. Otherwise, this statement is a pure hypothetical claim from the authors, unless the authors can provide analytical proof.

We agree with the Reviewer that our statement, without literature support, was purely hypothetical. We added references confirming the statement about crystal formation during combustion synthesis techniques into the Manuscript.

3. Line 75: Check the figure referencing of Figure S3.

We fixed the Figures and Tables referencing throughout the Manuscript.

4. Line 76: "Onion-like carbon structures" is not commonly used in the literature to describe amorphous carbon coating around the particle. The authors should elaborate more on this.

The carbon onions is a popular term in Nanotechnology concerning carbon structures (see <https://pubs.rsc.org/en/content/articlehtml/2016/ta/c5ta08295a>

<https://www.beilstein-journals.org/bjnano/articles/5/207>

<https://www.sciencedirect.com/science/article/pii/S0008622307004125?via%3Dihub>

<https://www.sciencedirect.com/science/article/pii/S037877531401355X?via%3Dihub>

<https://pubs.rsc.org/en/content/articlelanding/2013/ta/c3ta12628e>).

In the case of NVPF@C composite powder, we did not acquire carbon onion structures, however, the carbon discovered on the surface of NVPF particles resembled the carbon onions in some way. One can see a layered, round patterns on the HR-TEM images of NVPF@C composite powder (Figure S3), which we described as "onion-like" due to the resemblance with carbon onions. We added more HR-TEM images of NVPF@C composite powder into the Supplementary Information and added Figure S3 reference into the text.

5. Figure 4: The lines of both the plots can be thicker to enhance legibility and clarify of the data.

We thickened the lines connecting the points on the Figure 4.

6. Line 104: The removal of surface moisture during TGA will be complete at around 100 deg C. Only certain types of moisture (e.g. absorbed moisture or water molecules tightly associated with the material) may not fully evaporate until higher temperatures are reached. Therefore, the authors will need to justify why the mass loss up to 300 deg C represents loss of surface moisture.

We agree with the Reviewer about the removal of water above 300 deg. C during TGA experiments. Our statement was too great of a mental shortcut. We added the information about removal of water particles remaining in the crystal structure at temperatures above 100 deg. C and added a reference supporting that claim.

7. Line 115: Please elaborate the meaning of “concerning the mass of the whole NVPF@C composite”.

The NVPF@C composite powder is synthesised with a carbon coating around the NVPF particles present. Determination of specific capacity of NVPF@C powders included the presence of carbon in the powder (ca. 6.9% mas., as calculated from TGA experiment). We included additional information about carbon coating in the Manuscript.

8. In the main text, the result in Figure 6C is discussed before Figure 6B. This is not a serious issue, but to promote ease of reading, it is recommended to rearrange the figures.

We rearranged the figures on Figure 6.

9. Line 123: Two distinct voltage plateaus at 3.6 and 4.05 V are observed in the voltage profile of the synthesised NVPF. Since the authors have claimed that these plateaus do not correspond to pure NVPF (3.7 and 4.2 V), and there is no plateau at 3.4 V which corresponds to NVP, the reviewer fails to understand the significance of the plateaus at 3.6 and 4.05 V. There is a huge mismatch in position the voltage plateaus.

Due to the inductive effect caused by the fluorine anions present in the crystal matrix, the average working potential of NVPF is increased linearly with the fluorine content in $\text{Na}_3(\text{VO}_{1-x})_2(\text{PO}_4)_2\text{F}_{1+2x}$ ($0 \leq x \leq 1$) in the entire range of x . As the plateau potentials do not match with the fully fluorinated compound of $\text{Na}_3\text{V}_2(\text{PO}_4)_2\text{F}_3$, it suggests that fluorine content in our powder is lower than in $\text{Na}_3\text{V}_2(\text{PO}_4)_2\text{F}_3$ (which is also proven through XRD data and c -parameter calculations). Moreover, the average working potential of NVPF@C powder corresponds well with the literature data presented in Park et. al. (10.1002/adfm.201400561). We included additional information about working potentials of NVPF@C composite powder in the Manuscript.

10. Also to avoid any ambiguity of the origin of the observed voltage profiles, it is suggested that the authors conduct an ex-situ XPS analysis on the cycled NVPF electrode (refer to Section 3.6 of this paper: <https://www.sciencedirect.com/science/article/abs/pii/S2405829717301423>).

We cross-referenced the average working potential for our composite powder with the relationship between fluorine content and average working potential provided by Park et al. (10.1002/adfm.201400561) and found a very good correlation for powders synthesised by us through solution-combustion method.

As the main topic of the article is to present a new, never described before method of synthesising fluorinated compounds from $\text{Na}_3(\text{VO}_{1-x})_2(\text{PO}_4)_2\text{F}_{1+2x}$ ($0 \leq x \leq 1$) family and not to focus purely on the valence states present in our system, we decided to push the XPS analysis towards our future work, which will focus on optimisation of the solution-combustion method to obtain $\text{Na}_3(\text{VO}_{1-x})_2(\text{PO}_4)_2\text{F}_{1+2x}$ ($0 \leq x \leq 1$) compounds in the entire range of x . The XPS data will provide us with additional information about vanadium valence states in the entire range of NVPF fluorination, which later will be used to compare our data with previously described in the literature.

We added additional information about future plans regarding solution-combustion synthesis into the Manuscript.

11. Line 164: Instead of using “oxidised state”, the more commonly used term is “fully charged state”.

We replaced the “oxidised state” with “fully charged state” in the Manuscript.

12. The Na chemical diffusion coefficient can be easily determined from the low frequency EIS plot (Warburg contribution). Please refer to these papers: <https://link.springer.com/article/10.1007/s10008-011-1426-4> and <https://www.sciencedirect.com/science/article/abs/pii/S1388248108003469>.

We agree with the Reviewer, that sodium cation diffusion coefficient can be determined from the low frequency part of the EIS spectrum, as long as the Warburg contribution (45° inclined line) can be easily extracted from the complex-plane plots (which is the case of the papers suggested by the Reviewer and our previous work on lithium-titanium oxide: 10.1016/j.electacta.2016.10.018; 10.1039/C7RA10608D).

However, in the case of fast electron and ion transfer through the electrode, and/or thin layer or small particles present in the system, the mid to low frequency AC signal can penetrate into the boundary of the film or centre of the particle and, instead of producing a characteristic straight line inclined at 45° at complex-plane plots (originating from Warburg mass transfer), it gives almost vertical, capacitive response (the Na⁺ cannot penetrate deeper than the centre of the particle, resulting in capacitive-like response). Due to those effects, sodium cation transfer cannot be distinguished from the charge transfer through electrode/electrolyte interphase and Na⁺ diffusion coefficient cannot be extracted from the EIS data.

We added additional information and references about EIS data into the Manuscript.

13. Line 211: This section can be conveniently named Material Characterisations.

We changed the section heading to Material Characterisation.

14. Line 245: Double check if the electrode mass loading unit is g/cm² or mg/cm².

We corrected the typo to mg cm⁻².

15. Since the authors report a NVPF@C composite, it is recommended that a TGA is performed to quantify the amount of carbon present in the composite.

The TGA analysis have been performed and the results are shown on Figure 3. The amorphous carbon quantity was estimated to be ca. 6.9% mas.

16. Standardise the use of “Na⁺ ion”, “Na⁺ cation” and “Na⁺”. The correct representation should be either “Na⁺” or “Na ion”. Since Na⁺ already stands for sodium ion, writing Na⁺ ion is redundant.

We standardised the naming of sodium cations throughout the Manuscript.

17. The authors should review the language and tenses throughout the manuscript.

We reviewed the language and punctuations throughout the Manuscript.

Dear Reviewers,

we thank you for your feedback and the thorough review process of our manuscript.

With my best regards,

Dr Michal Krajewski on behalf of the authors.

Reviewer #1 (Remarks to the Author):

The authors have addressed the reviewer comments, and the article can now be accepted for publication.

Reviewer #2 (Remarks to the Author):

I have reviewed the changes and I am happy to recommend the publication of this manuscript.

Reviewer #3 (Remarks to the Author):

I have reviewed the authors' responses and the revised manuscript. I appreciate the thoughtful and thorough manner in which all comments and feedback are addressed.

The revisions have significantly improved the clarity and quality of the manuscript. Hence, I have no further concerns and am satisfied with the current version.